# Normalization of Vitamin D Serum Levels in Patients with Type Two Diabetes Mellitus Reduces Levels of Branched Chain Amino Acids

**DOI:** 10.3390/medicina58091267

**Published:** 2022-09-13

**Authors:** Mahmoud A. Alfaqih, Nebras Y. Melhem, Omar F. Khabour, Ahmed Al-Dwairi, Lina Elsalem, Tasnim G. Alsaqer, Mohammed Z. Allouh

**Affiliations:** 1Department of Physiology and Biochemistry, Faculty of Medicine, Jordan University of Science and Technology, Irbid 22110, Jordan; 2Department of Medical Laboratory Sciences, Faculty of Applied Medical Sciences, Jordan University of Science and Technology, Irbid 22110, Jordan; 3Department of Pharmacology, Faculty of Medicine, Jordan University of Science and Technology, Irbid 22110, Jordan; 4Department of Anatomy, Faculty of Medicine, Jordan University of Science and Technology, Irbid 22110, Jordan; 5Department of Anatomy, College of Medicine and Health Sciences, United Arab Emirates University, Al Ain 15551, United Arab Emirates

**Keywords:** branched-chain amino acids, diabetes, vitamin D, biomarkers, metabolism

## Abstract

*Background and Objectives*: Vitamin D is involved in pancreatic beta-cell function, insulin sensitivity, and inflammation. Further, elevation in branched-chain amino acids (BCAAs) has been implicated in type 2 diabetes (T2DM) pathology. However, the relationship between vitamin D and BCAAs in T2DM remains unclear. The current study aimed to investigate the relationship between vitamin D and BCAAs in T2DM. *Materials and Methods*: In total, 230 participants (137 with T2DM and 93 healthy controls) were recruited in a cross-sectional study. Furthermore, an additional follow-up study was performed, including 20 T2DM patients with vitamin D deficiency. These patients were prescribed weekly vitamin D tablets (50,000 IU) for three months. The levels of several biochemical parameters were examined at the end of the vitamin D supplementation. *Results*: The results showed that patients with T2DM had higher serum levels of BCAAs and lower serum levels of 25-hydroxyvitamin D (25(OH)D) compared with those of the healthy controls (*p* < 0.01). The serum levels of vitamin D were negatively correlated with BCAA levels in T2DM patients (r = −0.1731, *p* < 0.05). In the follow-up study, 25(OH)D levels were significantly improved (*p* < 0.001) following vitamin D supplementation. Vitamin D supplementation significantly reduced the levels of BCAAs, HbA1c, total cholesterol, triglycerides, and fasting glucose (*p* < 0.01). *Conclusion*: Overall, these results suggest a role for BCAAs and vitamin D in the etiology and progression of T2DM. Thus, managing vitamin D deficiency in patients with T2DM may improve glycemic control and lower BCAA levels.

## 1. Introduction

Diabetes mellitus (DM) is a growing crisis that challenges the public health sector and the global economy [1]. Type 2 DM (T2DM), caused by resistance to insulin activity, is the most common type of DM and is linked with obesity [2]. Despite measures adopted by many public health systems to reduce the risk of developing T2DM, the prevalence of this chronic disease is increasing, and T2DM is estimated to affect 578 million adults by 2030 [3]. T2DM is associated with many complications that significantly decrease the quality of life of affected patients [4]. Delayed wound healing, hypertension, atherosclerosis, and renal failure are among the many debilitating conditions that can be triggered or aggravated by T2DM [5]. However, our current understanding of T2DM indicates that such complications can be prevented and/or delayed by appropriate disease management.

T2DM is currently considered a heterogeneous group of disorders characterized by hyperglycemia [6]. T2DM primarily occurs because of resistance to insulin action in its target tissues, accompanied by a gradual loss of the ability of pancreatic tissues to compensate for the increasing demand for insulin secretion [7]. Obesity, a sedentary lifestyle, and dietary factors are commonly linked with the increasing prevalence of T2DM [8].

Isoleucine, leucine, and valine are three essential amino acids, and are collectively known as branched-chain amino acids (BCAAs) [9]. These amino acids are classified within the same group in view of similarities in their structural geometry and metabolism [10]. A link between BCAAs and the pathogenesis of DM was first observed by Felig et al. in 1970. In their report, patients with diabetes complicated by ketoacidosis showed increased BCAA levels in their blood [11]. Recent epidemiological evidence from several patient cohorts reinforces the presence of a relationship between elevated BCAA levels and T2DM [12,13,14,15]. Indeed, in a meta-analysis including seven population-based studies, circulating levels of BCAAs were found to increase the risk of T2DM by 40% [16].

BCAA levels are not only associated with the risk of developing T2DM, but are also correlated with several other clinical parameters that reflect glycemic control in these patients. For example, a positive association was found between circulating levels of BCAAs, with the Homeostatic Model Assessment for Insulin Resistance (HOMA-IR) score, and hemoglobin A1c (HbA1c) levels [17,18].

Lack of glycemic control in patients with T2DM is a well-established risk factor for the development of DM-related complications. Since T2DM patients with poor glycemic control have higher levels of BCAAs in their blood, it is assumed that higher BCAA levels could be an independent risk factor for DM-related complications. Several reports support the above hypothesis [19,20]. For example, Lim et al. demonstrated that blood BCAA levels were independently associated with heart failure incidents in a cohort of T2DM patients [21].

Based on the abovementioned reports, the adoption of therapeutic/dietary interventions that lead to lowering of BCAA blood levels in patients with T2DM may enhance glycemic control and/or reduce the risk of further complications. Along these lines, Karusheva et al., found that reducing dietary intake of BCAAs improves the metabolic profile of white adipose tissues and the gut microbiome content in patients with T2DM [22].

Vitamin D, or cholecalciferol, is a fat-soluble vitamin [23] that is important for maintaining calcium homeostasis and bone health [24]. Vitamin D is increasingly recognized for its anti-proliferative, pro-differentiative, and immunomodulatory activities [25,26]. In recent years, vitamin D has been shown to be involved in regulating insulin action on its target tissues [27,28].

An association between vitamin D and T2DM has been reported in several populations [29,30,31]. Furthermore, normalization of serum vitamin D levels in patients with T2DM who also have vitamin D deficiency significantly improved their glycemic control [32,33,34]. Recently, vitamin D has been shown to selectively enhance the catabolism of BCAAs in monocytes [35]. To the best of our knowledge, the relationship between vitamin D and BCAAs in T2DM has not been thoroughly investigated.

Therefore, in this study, we examined the hypothesis that serum levels of vitamin D are negatively correlated with serum BCAA levels in patients with T2DM, and that normalization of serum vitamin D levels using approved supplementation protocols can lower serum BCAA levels in these patients.

## 2. Materials and Methods

### 2.1. Study Design

A case-control design was used in this investigation, succeeded by an interventional follow-up study, as summarized in Figure 1. Approval to recruit participants was granted by the Institutional Review Board at the Jordan University of Science and Technology (JUST, IRB #52/135/2020). Informed consent was obtained from all subjects involved in the study. Consent was collected at baseline, before the case-control part, and again before the vitamin D intervention part. Participants were recruited between June and September 2021 at the outpatient clinics of King Abdullah University Hospital (KAUH), a tertiary hospital affiliated with JUST.

Two hundred and thirty participants were recruited for the case-control part of this study. The number of patients diagnosed with T2DM was 137, whereas the number of disease-free controls was 93. T2DM was diagnosed according to the American Diabetes Association (ADA) guidelines. All patients with T2DM were recruited from the endocrinology clinics of KAUH. These patients were actively treated for T2DM at the time of recruitment. Patients were included in the T2DM group if they were clinically diagnosed with T2DM, regardless of their HbA1c levels at the time of recruitment.

The control arm included 93 T2DM-free participants recruited from the family medicine clinics of KAUH. The absence of T2DM in the control arm of the study was assessed by a clinical research coordinator prior to patient recruitment. Initially, the clinical research coordinator interviewed the participant and assessed the presence or absence of any symptoms associated with a potential T2DM diagnosis (i.e., increased thirst, frequent urination, and frequent infections). Potential control subjects who did not report any of these symptoms were requested to visit the clinic for a future blood sampling. During the subsequent visit, 5 mL of whole blood was collected from the participant into an ethylene-diamine-tetra-acetic acid (EDTA) tube (AFCO, Jordan), and the sample was submitted to the laboratories of KAUH for hemoglobin A1c (HbA1c) measurement.

Participants were included in the control arm of the study if they had an HbA1c of 5.6% or less and were not clinically diagnosed with T2DM or prediabetes. Subjects were excluded from the control arm if they had an HbA1c measurement of 5.7 or more.

Patients were excluded from both arms of the case-control study if they indicated that they took vitamin D supplements, in any form, within the last three months before recruitment. Patients with chronic disease conditions known to affect vitamin D absorption or its metabolism, as well as those with a history of allergy to any of the vitamin D pharmacological preparations, were also excluded from the study. All participants included in the study were of Jordanian descent.

### 2.2. Vitamin D Intervention

Of the 137 patients with T2DM, 122 had 25(OH) vitamin D levels lower than 20 ng/mL, and were thus diagnosed with vitamin D deficiency according to the Endocrine Society clinical practice guidelines [36]. Ten patients were diagnosed with vitamin D insufficiency (20–29 ng/mL), while five T2DM patients only had normal vitamin D levels (>30 ng/mL). All patients with vitamin D deficiency or insufficiency were invited to participate in the interventional follow-up study. However, only twenty patients with vitamin D deficiency consented to participate. These patients were prescribed weekly (50,000 IU) vitamin D3 tablets (Biodal, Amman, Jordan), administered orally for three months. The above dosing regimen matches the Endocrine Society clinical practice guidelines [36]. The patients were monitored by a clinical research coordinator for the duration of the vitamin D supplementation to report the presence of any potential side effects. All patients enrolled in the intervention group were requested to maintain their T2DM management protocol (i.e., dietary habits, drug intake, exercise frequency and intensity, and weight control measures) for the entire duration of the follow-up study.

At the end of the intervention period, patients were requested to fast for 12 h and visit their respective physician in the morning. During the visit, two blood samples (5 mL each) were sampled from each patient by a certified phlebotomist. One blood sample was collected into an ethylene-diamine-tetra-acetic acid (EDTA) tube (AFCO, Amman, Jordan) and later used for HbA1c measurement performed at the laboratories of KAUH. The other sample was collected into a plain tube with a gel clot activator (AFCO, Jordan) for serum separation, after centrifugation for 5 min at 4000× *g*.

Serum samples were transferred to 1.5 mL microcentrifuge tubes (Eppendorf) and stored at −80 °C until further use. Serum collected from the plain tubes was used to measure the levels of 25-hydroxyvitamin D (25(OH) vitamin D), glucose, total cholesterol, triglycerides, and BCAAs as described in the following sections.

### 2.3. Anthropometric Measurements

Height (cm), weight (kg), and waist circumference (WC) (cm) were measured for all participants in the study. In the intervention group, height, weight, and WC measurements were repeated on the day of post-supplementation blood sampling (i.e., one week after receiving the last dose of vitamin D3), in addition to the measurements made at the baseline. Body mass index (BMI) was calculated as weight/height^2^ (kg/m^2^). Waist circumference (WC) was measured using a tape measure positioned at the midpoint between the inferior costal margin and the superior border of the iliac crest.

### 2.4. HbA1c Measurement

HbA1c measurement was performed at the Clinical Chemistry laboratories of KAUH, using an automated analyzer system (Roche Diagnostics, Mannheim, Germany).

### 2.5. Biochemical Measurements

Fasting glucose, total cholesterol, triglycerides, 25(OH) vitamin D, and BCAAs were measured in the serum samples collected from the participants of the case-control study at the baseline. They were also measured in the serum samples collected from patients who participated in the vitamin D intervention group after one week of receiving the last dose of vitamin D3.

The serum levels of BCAA, glucose, total cholesterol, and triglycerides were analyzed using commercially available colorimetric assay kits purchased from Abcam (Cambridge, MA, USA) according to the manufacturer’s instructions. The serum levels of 25(OH) vitamin D were measured using an ELISA-based kit purchased from Abcam (Cambridge, MA, USA) according to the manufacturer’s instructions.

### 2.6. Statistical Analysis

Data were analyzed using SPSS software version 22 (IBM SPSS Inc., Chicago, IL, USA). Continuous variables were expressed as mean ± standard deviation. Statistical differences between patients with diabetes and the controls were tested using a two-sided Student’s *t*-test for the following continuous variables: age, body mass index (BMI), waist circumference (WC), cholesterol, glucose, triglycerides, HbA1c, 25(OH) vitamin D, and BCAAs. Differences in gender distribution between patients and their controls were determined using Pearson’s chi-square test. A paired *t*-test was used to test the presence of statistically significant differences in 25(OH) vitamin D levels of participants grouped according to normal and high levels of triglycerides, glucose, HbA1c, or cholesterol. A paired *t*-test was also used to determine statistically significant differences in cholesterol, glucose, triglyceride, HbA1c, 25(OH) vitamin D, and BCAA levels among the patients prior to vitamin D supplementation, compared with the measurements made following the intervention. Pearson’s correlation coefficient was used as a measure of association. Statistical significance was set at *p* < 0.05.

## 3. Results

### 3.1. Baseline Characteristics

In total, 230 participants met the eligibility criteria and consented to participate in this study. The number of participants diagnosed with T2DM was 137, while the number of controls was 93. Table 1 illustrates the general anthropometric and biochemical characteristics of the participants. Data are presented as the mean ± SD. Participants with T2DM were slightly older (*p* < 0.05) and had a slightly higher BMI (*p* < 0.01) and waist circumference (*p* < 0.01) compared to the BMI and WC measurements of the healthy controls.

Evaluation of several biochemical parameters indicated that participants with T2DM had significantly higher levels of glucose (*p* < 0.01), HbA1c (*p* < 0.01), and BCAAs (*p* < 0.01) compared with those in the control group. Moreover, the patients with T2DM had significantly (*p* < 0.01) lower levels of serum 25(OH) vitamin D (Table 1).

Differences observed in BCAAs and 25(OH) vitamin D levels between diabetic cases and controls could be caused by variations in age, BMI, or WC. Therefore, a logistic regression analysis was performed to confirm the association of serum BCAAs and 25(OH) vitamin D with T2DM. In this analysis, it was observed that serum BCAAs significantly increased the risk of T2DM, while 25(OH) vitamin D reduced the risk (*p* < 0.01) (Table 2).

### 3.2. Relationship between Serum 25(OH) Vitamin D and Biochemical Markers

We categorized the study participants into high versus normal levels of glucose, HbA1c, triglycerides, and cholesterol based on the ACC-AHA approved guidelines. We then compared the serum levels of 25(OH) vitamin D between the high versus normal groups of each of the aforementioned biochemical markers. Interestingly, serum 25(OH) vitamin D levels were significantly lower in participants categorized within the high group of triglycerides (*p* < 0.05), HbA1c (*p* < 0.01), and glucose (*p* < 0.01), as shown in Figure 2B, 2C, and 2D, respectively. However, there was no significant difference in 25(OH) vitamin D levels between participants with high versus normal cholesterol (Figure 2A).

### 3.3. Correlation between Serum Levels of BCAAs and 25(OH) Vitamin D

In this part, we first investigated the relationship between serum BCAA levels and HbA1c. Significantly higher levels of serum BCAAs (*p* < 0.01) were found in participants who had a HbA1c level higher than 5.6% compared to participants with lower HbA1c (Figure 3).

In the second part, linear regression was used to investigate if a correlation was present between serum 25(OH) vitamin D and BCAAs among T2DM patients. In this analysis, it was determined that a significantly negative correlation existed between the two above biochemical markers (r = −0.1731, *p* < 0.05), where lower levels of serum 25(OH) vitamin D were associated with higher levels of serum BCAAs in T2DM patients (Figure 4).

### 3.4. Vitamin D Intervention

Twenty patients with T2DM and vitamin D deficiency consented to participate in this interventional follow-up study. Changes in several biochemical measures were found at the end of the vitamin D intervention period (Table 3). Serum 25(OH) vitamin D levels after supplementation were significantly higher than those prior to supplementation (*p* < 0.01), confirming the treatment efficacy. Moreover, there was a significant reduction in BCAA levels after vitamin D supplementation (*p* < 0.01). Interestingly, vitamin D supplementation also significantly reduced the serum levels of HbA1c (*p* < 0.05), cholesterol (*p* < 0.01), triglycerides (*p* < 0.01), and glucose (*p* < 0.05).

## 4. Discussion

In this investigation, relationships between the serum levels of vitamin D and several other biochemical markers (HbA1c, glucose, cholesterol, triglycerides, and BCAAs) were investigated in a cross-section of patients with T2DM and disease-free controls. This assessment was followed by an interventional follow-up study in which we used (1) a clinically approved vitamin D supplementation protocol to normalize the serum levels of 25 (OH) vitamin D, and (2) assessed the effect of this protocol on the serum levels of the same set of biochemical markers that were under investigation in the first part of the study.

In the first part of the study, serum levels of 25(OH) vitamin D were found to be significantly lower in patients with T2DM. Furthermore, lower levels of serum 25(OH) vitamin D were found to be associated with higher levels of HbA1c, glucose, and triglycerides. Moreover, study participants (T2DM patients and their controls) with high HbA1c (indicated by HbA1c levels of 5.6% or higher) were found to have higher serum levels of BCAAs. Finally, a negative correlation was observed between the serum levels of 25(OH) vitamin D and serum BCAAs in patients with T2DM.

In the second part of this investigation, we demonstrated that normalization of 25(OH) vitamin D serum levels in patients with T2DM who were diagnosed with vitamin D deficiency is accompanied by a concomitant reduction in the serum levels of glucose, triglycerides, and BCAAs. Further, glycemic control in these patients was improved, as indicated by a significant reduction in HbA1c levels.

The findings of this report showed that BCAAs were elevated in T2DM, and that this elevation was more pronounced in patients with poor glycemic control, as indicated by higher BCAA levels in participants with higher HbA1c levels. High-protein diets have been shown to be associated with impaired glucose tolerance, insulin resistance, and an increased incidence of T2DM [37]. As BCAAs are the most abundant amino acids in proteins, this may explain the higher risk of T2DM among individuals who consume high protein diets [38].

The observed elevation of BCAAs among patients with T2DM reported in the present study confirmed the role of BCAAs in the disease etiology. The results also showed a positive linear relationship between the levels of BCAAs and HbA1c. These results are consistent with previous investigations conducted in other populations [38,39,40,41,42].

Hypertension, which is usually present alongside T2DM [43], is linearly associated with an increased level of BCAAs [44,45]. Moreover, increased levels of BCAAs were found to be associated with incident heart failure in patients with T2DM [21]. Several studies have revealed that higher levels of BCAAs are associated with dyslipidemia and cardiovascular events [19,46,47,48]. Thus, increased BCAA levels seem to be a central component in T2DM pathology and its complications.

Interestingly, reducing BCAA levels has been shown to improve overall metabolic health and ameliorate T2DM complications [13,49,50]. Yun et al. found that a diet low in BCAAs was associated with lower HbA1c levels [51]. Further, reduction of dietary BCAAs was found to decrease postprandial insulin secretion and improve white adipose tissue metabolism and gut microbiome composition in patients with T2DM [22]. Thus, it appears that interventions that reduce BCAAs can significantly improve glycemic control in T2DM, and may potentially protect the patients from associated complications.

Our observation that a decrease in serum BCAA levels can be achieved via normalization of vitamin D serum levels highlights vitamin D supplementation as an effective protocol for managing T2DM and lowering the risk of future complications of this metabolic disease.

In this report, it was revealed that supplementation with vitamin D reduces serum BCAA levels. The exact molecular mechanism that explains this finding requires further investigations. However, it could be attributed to an increase in BCAA catabolism induced by 25(OH) vitamin D. Dimitrov et al. observed that treatment of monocytic cells with 1,25(OH) vitamin D up-regulates the expression of cytoplasmic branched-chain aminotransferase (BCAT1) gene and genes which encode components of branched ketoacid dehydrogenase complex (BCKDH) [35]. Interestingly, these genes are involved in BCAA catabolism, and could indicate that vitamin D would enhance BCAA degradation, eventually leading to a decrease in serum levels of BCAAs. However, this hypothesis still needs formal testing in relevant cell lines and animal models.

Furthermore, evidence supports that vitamin D could indirectly attenuate the loss of insulin sensitivity induced by BCAAs. In this context, it is broadly accepted that BCAAs inhibit insulin sensitivity by activating the mammalian target of rapamycin complex 1 (mTORC1) [52]. Interestingly, pretreatment with 1,25(OH) vitamin D suppressed BCAA-dependent activation of mTOR signaling in monocytic cells [35]. This finding indicates that vitamin D may also mechanistically attenuate the BCAA-inhibiting effect on insulin sensitivity.

A disturbing finding of this report was the low levels of vitamin D among the participants. Indeed, this report revealed that the mean values of serum 25(OH) vitamin D in both T2DM cases and disease-free controls were lower than the cut-off value for vitamin D deficiency (i.e., 20 ng/mL) [53].

The sample size and geographic distribution of the patients included in this report do not represent the population in Jordan. However, our findings concur with other, more comprehensive reports, which assessed the levels of 25 (OH) vitamin D across different age groups. For example, in a representative sample that included 4056 participants older than 17 years, El-Khateeb et al., reported the overall prevalence of vitamin D deficiency to be 89.7% [54]. Moreover, Abdul-Razzak et al., demonstrated the prevalence of vitamin D deficiency to be 29% in a cross-sectional sample including 275 healthy infants and toddlers aged between 6 to 36 months [55].

Low levels of vitamin D increase the risk of T2DM, as well as the risk of cancer [56], infertility [57], and metabolic syndrome [58]. The findings of this report, which associate lower levels of vitamin D with a higher risk of T2DM in Jordan, in addition to the body of literature reporting a high prevalence of vitamin D deficiency in Jordan, highlight the need for this issue to be addressed by the Jordanian public health authorities.

Indeed, there is an eminent need to initiate nationwide awareness campaigns that explain the dietary and environmental sources of vitamin D, the link of vitamin D deficiency with chronic diseases, and the relative safety and cost-effectiveness of vitamin D supplementation protocols in preventing deficiency, along with its numerous public health implications.

Although the patients enrolled in the supplementation program were instructed to maintain their dietary and exercise routine for the duration of the intervention, the reduction observed in the serum levels of BCAAs following vitamin D supplementation could be caused by a change in the eating behavior of the study participants. This confounding factor could have been avoided by collecting information on the dietary intake of BCAAs using food frequency questionnaires [59]. Another limitation was the small sample size of the intervention group and the fact that all patients were from a single institution situated in the Northern part of Jordan. A third limitation of this report was the absence of a control group of T2DM patients with vitamin D deficiency who did not receive vitamin D supplementation and received a placebo instead. The inclusion of such a group would have provided additional evidence that the changes observed in BCAA serum levels are due to vitamin D supplementation and are not due to changes in any other potential confounding variable.

## 5. Conclusions

In conclusion, this study is the first to show that BCAA levels in the serum of patients with T2DM can be decreased through a clinically approved vitamin D supplementation protocol which effectively normalizes 25(OH) vitamin D in patients with T2DM diagnosed with vitamin D deficiency. This finding could have implications on the management of and/or the prevention of future complications of T2DM.

## Figures and Tables

**Figure 1 medicina-58-01267-f001:**
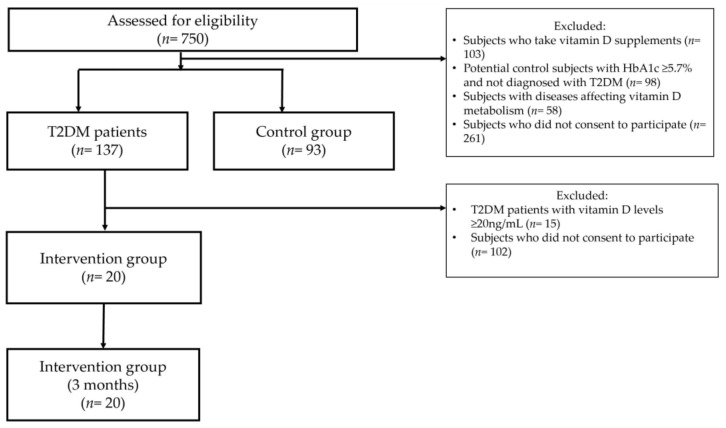
Flow chart summarizing the study design with relevant exclusion criteria.

**Figure 2 medicina-58-01267-f002:**
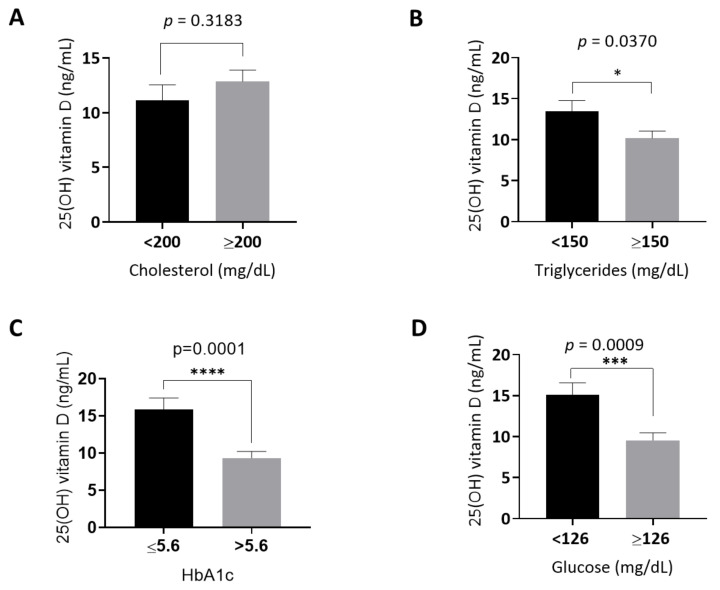
Comparison of serum 25(OH) vitamin D levels in participants who had high versus normal levels of (**A**) cholesterol, (**B**) triglycerides, (**C**) HbA1c, and (**D**) glucose.

**Figure 3 medicina-58-01267-f003:**
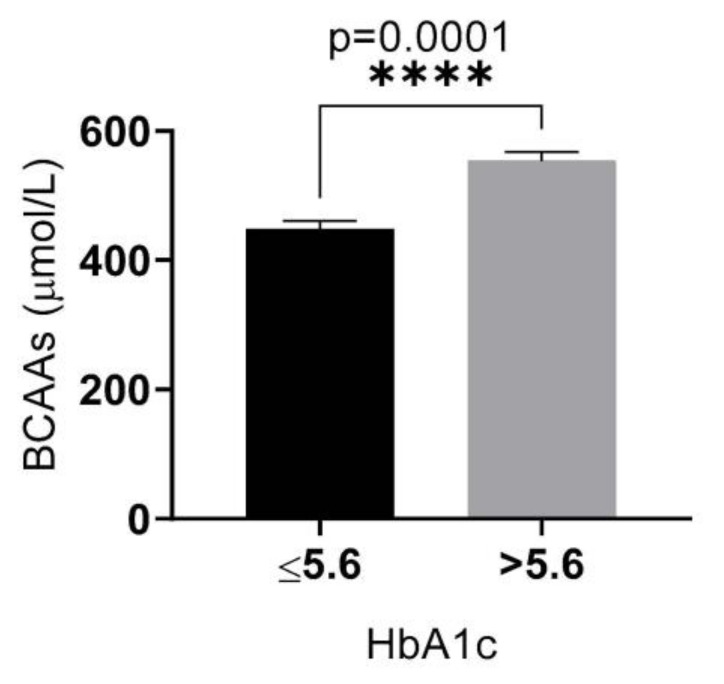
Comparison of serum BCAA levels in the study population according to HbA1c level.

**Figure 4 medicina-58-01267-f004:**
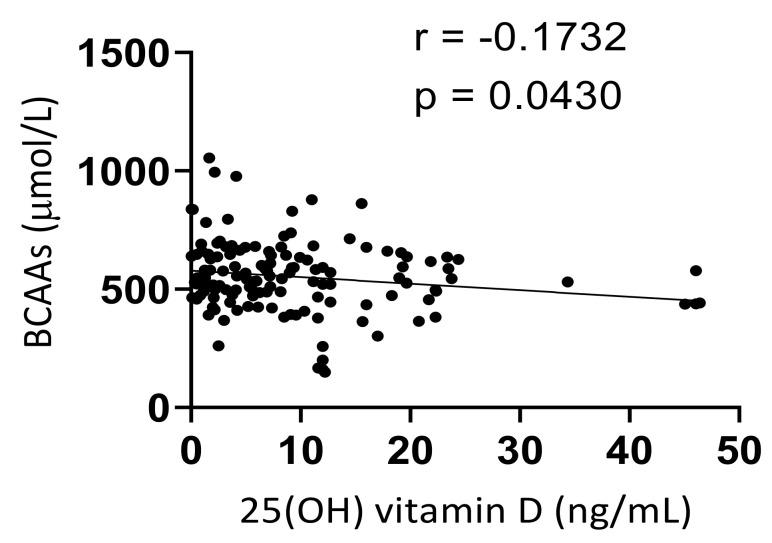
Correlation between serum levels of 25 (OH) vitamin D (ng/mL) and BCAAs (μmol/L) in patients with T2DM. r represents the Spearman’s correlation coefficient.

**Table 1 medicina-58-01267-t001:** Baseline variables of the study participants.

Variable	Controls *n* = 93	Diabetic *n* = 137	*p*-Value ^1^
Age (years)	51.86 ± 9.40	55.03 ± 9.40	0.013
Gender, *n* (%)			
Male	34 (36.5%)	61 (44.5%)	0.228 ^2^
Female	59 (63.4%)	76 (55.4%)	
(BMI) (Kg/m^2^)	29.96 ± 4.87	32.81 ± 5.33	<0.0001
WC (cm)	98.97 ± 12.88	111.51 ± 12.10	<0.0001
Cholesterol (mg/dL)	219.58 + 44.65	215.12 ± 60.41	0.539
Triglycerides (mg/dL)	150.47 ± 124.98	174.04 ± 99.41	0.112
Glucose (mg/dL)	100.75 ± 18.45	204.28 ± 82.22	<0.0001
HbA1c	5.31 ± 0.26	8.25 ± 1.97	<0.0001
25(OH) vitamin D (ng/mL)	16.4 ± 15.39	9.29 ± 9.62	<0.0001
BCAAs (µmol/L)	443.27 ± 122.82	553.14 ± 148.97	<0.0001

^1^ The *p*-values were calculated using the student’s *t*-test, except for gender. ^2^ the *p*-value for gender was calculated using Pearson’s χ^2^ test of association. The data are presented as the mean ± standard deviation. Abbreviations: BMI, body mass index; WC, waist circumference; HbA1c, glycated hemoglobin; BCAAs, Branched chain amino acids; *n*, number.

**Table 2 medicina-58-01267-t002:** Logistic regression analysis of the study variables.

Variable	Odds Ratio	CI (95%)	*p*-Value
Age (years)	1.039	1.005–1.075	0.025
Gender, (female/male)	0.612	0.306–1.225	0.165
(BMI) (Kg/m^2^)	1.095	1.026–1.168	0.006
Cholesterol (mg/dL)	1.000	0.994–1.006	0.880
Triglycerides (mg/dL)	1.000	0.997–1.003	0.888
25(OH) vitamin D (ng/mL)	0.957	0.932–0.983	0.001
BCAAs (µmol/L)	1.006	1.003–1.009	0.000

Abbreviations: BMI, body mass index; BCAAs, branched chain amino acids; CI, confidence interval.

**Table 3 medicina-58-01267-t003:** Vitamin D supplementation.

Variable	Pre-Supplementation *n* = 20	Post-Supplementation *n* = 20	*p*-Value
HbA1c	8.35 ± 1.96	7.48 ± 1.45	0.0135
25(OH) Vitamin D (ng/mL)	1.26 ± 0.78	33.60 ± 18.06	<0.0001
BCAAs (µmol/L)	457.15 ± 139.63	338.86 ± 101.57	<0.0001
Cholesterol (mg/dL)	196.39 ± 46.26	74.99 ± 15.72	0.0001
Triglycerides (mg/dL)	178.77 ± 117.23	128.83 ± 88.44	0.0041
Glucose (mg/dL)	212.68 ± 75.13	186.11 ± 71.19	0.0480

The *p*-values were calculated using the student’s *t*-test. The data are presented as the mean ± standard deviation. Abbreviations: HbA1c, glycated hemoglobin; BCAAs, Branched-chain amino acids; *n*, number.

## Data Availability

The datasets generated and/or analyzed during the current study are available from the first or corresponding author on reasonable request.

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
