# Peer review of "Normalization of Vitamin D Serum Levels in Patients with Type Two Diabetes Mellitus Reduces Levels of Branched Chain Amino Acids"

_medicina, 2022, doi:10.3390/medicina58091267_

Round 1
Reviewer 1 Report
The study entitled „Normalization of Vitamin D Serum Levels in Patients with Type Two Diabetes Reduces Levels of Branched Chain Amino Acids” by Alfaqih et. Al, presents two studies in one, where they first used a case-control design to determine if there 230 patients (137 with T2DM and 93 controls) would show a relationship between the serum 25(OH) vitamin and biomarkers.
Further down they discovered a negative correlation between branch chained amino acids (BCAAs) and 25 (OH) vitamin in the first study group. Thus they hypothesized, on the observation that vitamin D 25(OH) lowers BCAAs in T2DM patients. So they tried thre hypothesize in a second part of the study where they used 20 participants from the initial study group with T2DM and hypovitaminosis D and prescribed oral tables of 50000 IU of vitamin D3 once per week for 3 months.
The paper is overall well-conceived and written and contains an interesting topic which is in my opinion suitable and in the scope of your journal.
Selection of the study population was well thought out, but all participants in the study were only from one country.
Hower some issues should be addressed:
Principle concerns:
- For the intervention protocol, it would have been a better approach to add an additional placebo control group.
Major concerns:
- In Figure 1C & 2: Why is the HbA1c group “>5.6” when you did not include patients with an HbA1c from 5.7 – 6.4 (prediabetes). Was >5.6 the lowest HbA1c from a Patient that was clinically diagnosed with T2DM? Wouldn’t >=6.5 be the appropriate group? What is the explanation behind that? Please explain it in more detail in the manuscript.
- Did the intervention group knew that they were in an intervention group? Were eating habits monitored in the 3 months? How do you explain now the sudden significantly reduction of cholesterol post supplementation of Vtamin D.
- Show the correlation of Vitamin D and BCAA again in the intervention group.
Minor concerns:
- Line 274 “..lower levels of serum 25(0H)” should be (OH)
Reviewer 2 Report
The subject has enough novelty. But there are some limitations that may affect the results.
Introduction
1. The manuscript needs to improve in terms of scientific writing and grammatical points. A native edition is suggested.
2. Lines 70 and 71 are not clear.
Materials and methods
1. It seems necessary to present the study design and the recruiting of the participants in a schematic algorithm.
2. As a clinical trial, it is necessary to state the registration number in addition to the approval number of the research.
3. This research is a cross-sectional design with an experimental study to evaluate the effect of vitamin D supplementation on the BCAA serum levels and lipid profile. However, research with this study design couldn’t be nominated as COHORT which is a study with ample sample size, long period of time to follow-up, specific exposure, and outcome, and conducted in a particular population.
4. The number of controls is usually more than the cases. It is better that the number of controls is 4 times more than the cases. However, in this search, the number of patients (cases) was nearly 1.5 times more than the number of healthy subjects (controls). How do researchers explain this limitation?
5. The word “Type 2 diabetes mellitus” used in this manuscript is applied in various terms, T2DM, DM, diabetes, … . Please equalize these terms throughout the manuscript.
6. The materials and method section should clarify the inclusion and exclusion criteria.
7. Line 110, what is meant by blood withdrawal? Also, withdrawn in line 111? If you mean blood sampling, please correct them.
8. Lines 122-123 are not clear. Please check them.
9. Lines 127-128, which serum levels of vitamin D were being diagnosed as vitamin D deficiency? Please insert the exact reference.
10. Why did the researchers exclude the DM patients with HbA1c levels of 6.5% or higher from the study?
11. Please give a reference for the dosage and duration of usage of vitamin D supplementation.
12. Did the researcher exclude the DM patients with vitamin D insufficiency from the cohort arm of the study? Why or why not?
13. Line 137, I think fasting for 15 hours is too long for DM patients. Fasting for 8 to 12 hours is enough to measure fasting blood sugar. Please explain the cause of this long fasting.
Results
1. “Vitamin D intervention”, it is better to move this section to the “materials and method” section. Please give a suitable reference for this section.
2. It is necessary to control the dietary vitamin D and BCAA intake and other dietary factors which may be affected the vitamin D and BCAA serum levels, and serum lipid profiles. Among these factors, we can mention meat intake. How the researcher controls these confounders in their study? How are they sure that the results obtained are only due to the effect of vitamin D supplementation?
Discussion
1. A suggested biochemical mechanism to discuss the relationship between BCAA and vitamin D as well as a mechanism to explain a pathophysiologic effect of the BCAA in diabetes mellitus and how vitamin D works in this pathology should be mentioned in the discussion section.
Round 2
Reviewer 1 Report
The authors sufficiently adressed all my comments.
The manuscript is no ready to be published.
Reviewer 2 Report
The manuscript entitled “Normalization of Vitamin D Serum Levels in Patients with Type Two Diabetes Mellitus Reduces Levels of Branched Chain Amino Acids” has been improved grammatically and scientifically. However, some points have been remained to consider.
1. The authors acknowledged that they highlighted the corrections in the text of the manuscript in yellow, but there is no highlighted section in the revised manuscript.
2. The number of the excluded subject should be stated in figure 1.
3. The causes stated in comment 2 about the approval number of the research are not acceptable. All clinical trials should be registered in the https://clinicaltrials.gov/
